# First Trimester Combined Test (FTCT) as a Predictor of Gestational Diabetes Mellitus

**DOI:** 10.3390/ijerph16193654

**Published:** 2019-09-28

**Authors:** Federica Visconti, Paola Quaresima, Eusebio Chiefari, Patrizia Caroleo, Biagio Arcidiacono, Luigi Puccio, Maria Mirabelli, Daniela P. Foti, Costantino Di Carlo, Raffaella Vero, Antonio Brunetti

**Affiliations:** 1Unit of Obstetrics and Gynecology, Department of Medical and Surgical Sciences, University “Magna Græcia” of Catanzaro, Viale Europa, 88100 Catanzaro, Italy; fed.visconti@gmail.com (F.V.); dr.paolaquaresima@gmail.com (P.Q.); cdicarlo@unicz.it (C.D.C.); 2Department of Health Sciences, University “Magna Græcia” of Catanzaro, 88100 Catanzaro, Italy; echiefari@libero.it (E.C.); arcidiaconob@gmail.com (B.A.); maria.mirabelli@unicz.it (M.M.); foti@unicz.it (D.P.F.); 3Complex Operative Structure Endocrinology-Diabetology, Hospital Pugliese-Ciaccio, 88100 Catanzaro, Italy; patrizia.caroleo@alice.it (P.C.); puccio55@libero.it (L.P.); rafvero@libero.it (R.V.)

**Keywords:** β-human chorionic gonadotropin, fetal nuchal translucency, first trimester combined test, gestational diabetes mellitus, pregnancy-associated plasma protein A (PAPP-A)

## Abstract

*Background*—The first trimester combined test (FTCT) is an effective screening tool to estimate the risk of fetal aneuploidy. It is obtained by the combination of maternal age, ultrasound fetal nuchal translucency (NT) measurement, and the maternal serum markers free β-human chorionic gonadotropin (β-hCG) and pregnancy-associated plasma protein A (PAPP-A). However, conflicting data have been reported about the association of FTCT, β-hCG, or PAPP-A with the subsequent diagnosis of gestational diabetes mellitus (GDM). *Research design and methods*—2410 consecutive singleton pregnant women were retrospectively enrolled in Calabria, Southern Italy. All participants underwent examinations for FTCT at 11–13 weeks (plus 6 days) of gestation, and screening for GDM at 16–18 and/or 24–28 weeks of gestation, in accordance with current Italian guidelines and the International Association Diabetes Pregnancy Study Groups (IADPSG) glycemic cut-offs. Data were examined by univariate and logistic regression analyses. *Results*—1814 (75.3%) pregnant women were normal glucose tolerant, while 596 (24.7%) were diagnosed with GDM. Spearman univariate analysis demonstrated a correlation between FTCT values and subsequent GDM diagnosis (*ρ* = 0.048, *p* = 0.018). The logistic regression analysis showed that women with a FTCT <1:10000 had a major GDM risk (*p* = 0.016), similar to women with a PAPP-A <1 multiple of the expected normal median (MoM, *p* = 0.014). Conversely, women with β-hCG ≥2.0 MoM had a reduced risk of GDM (*p* = 0.014). *Conclusions*—Our findings indicate that GDM susceptibility increases with fetal aneuploidy risk, and that FTCT and its related maternal serum parameters can be used as early predictors of GDM.

## 1. Introduction

Gestational diabetes mellitus (GDM) is one of the most common medical complications of pregnancy and represents a major risk factor for both adverse materno-fetal outcomes [1,2] and long-term maternal complications [3,4]. In recent years, its incidence has dramatically increased worldwide, in parallel with the rising incidence of overweight/obesity and common type 2 diabetes mellitus (T2DM) [1]. In 2010, the International Association of Diabetes and Pregnancy Study Group (IADPSG) panel recommended a universal screening, consisting of a 75 g oral glucose tolerance test (OGTT) to be performed at 24–28 weeks of gestation in all pregnant women, with lower glycemic cut-offs for the diagnosis of GDM with respect to the past [5]. The adoption of the IADPSG’s recommendations has widely increased the number of pregnant women diagnosed with GDM [1], and this has had a huge financial impact on the public healthcare system. For this reason, different screening criteria have been proposed and adopted worldwide, including a selective screening tool and different glycemic cut-offs [1].

The first trimester combined test (FTCT) identifies a population of women whose fetuses are at increased risk for trisomy 21 and other aneuploidy, with a detection rate of approximately 90% and a false-positive rate of 5% [6]. FTCT is performed between 11 and 13 weeks (plus 6 days) of gestation, and it is obtained by the combination of maternal age, ultrasound fetal nuchal translucency (NT) measurement, and the maternal serum markers free β-human chorionic gonadotropin (β-hCG) and pregnancy-associated plasma protein A (PAPP-A) [6]. NT refers to a fluid-filled subcutaneous space present in all fetuses, posterior to the cervical spine, physiologically varying from 0.7 mm at 10 weeks of gestation to 1.5 mm at 13 weeks [7]. Increased NT measurements are significantly associated with aneuploidy and structural malformations [7,8]. β-hCG is a pregnancy-specific hormone produced by trophoblast cells, which regulates placental development [9]. PAPP-A is a proteolytic enzyme produced by the placenta and decidua and is believed to have a critical function in the normal placental development. Abnormal concentrations of either β-hCG or PAPP-A are associated with adverse pregnancy outcomes, such as pre-eclampsia, pre-term delivery, spontaneous fetal loss, low birth weight, and small for gestational age [10,11,12,13,14,15]. In addition, a significant reduction in PAPP-A and free β-hCG concentrations has been reported in pregnant women that were subsequently diagnosed with GDM [11,16,17], while in a more recent study, it was suggested that high free β-hCG levels in the first trimester of pregnancy decrease the risk for GDM [18]. On the contrary, in two other studies [19,20], no significant association was observed between GDM and both markers.

In light of the above-mentioned conflicting reports, we herein examined the association of free β-hCG and PAPP-A serum levels with GDM in a large single-center population of pregnant women from Calabria, Southern Italy, which is a region of comparatively limited genetic diversity [21,22].

## 2. Materials and Methods

### 2.1. Study Population

This retrospective population-based study involved 2410 consecutive singleton pregnant women, attending the Operative Unit of Diabetes (Hospital Pugliese-Ciaccio, Catanzaro, Italy) for GDM screening, during the period from August 2011 to December 2016. All these women had previously undergone FTCT screening in the same outpatient clinic. Nearly all subjects (99.4%) were Caucasian women from Calabria, Southern Italy. Screening for GDM was performed using a 75 g oral glucose tolerance test (OGTT) at 16–18 weeks and/or 24–28 weeks of gestation, according to the Italian guidelines [23]. Diagnosis of GDM was made in accordance with the IADPSG glycemic cut-off values (fasting value ≥ 92 mg/dL (5.1 mmol/L), 1 h post-glucose load ≥ 180 mg/dL (10 mmol/L), 2 h post-glucose load ≥ 153 mg/dL (8.5 mmol/L)) [5]. Gestational age was confirmed by ultrasonography examination. Anamnestic information included age, ethnicity, parity, previous GDM, family history of diabetes (first- or second-degree relatives), preexisting polycystic ovary syndrome (PCOS) as defined by “The Rotterdam ESHRE/ASRM-sponsored PCOS consensus workshop group” criteria [24], smoking status, self-reported pre-pregnancy weight, FPG at pre-pregnancy or first pregnancy visit between 100 and 125 mg/dL (5.6–6.9 mmol/L), and previous macrosomia. Women with any form of preexisting diabetes mellitus, as defined by the American Diabetes Association (ADA) criteria, with multifetal gestation, untreated endocrinopathies, or active chronic systemic diseases, or women using medications that could affect glucose tolerance, were excluded from the study. Data collection was approved by the ethics committee of Regione Calabria Sezione Area Centro. As the data were analyzed anonymously, there was no need for written informed consent. The study was performed in accordance with the Declaration of Helsinki.

### 2.2. Ultrasound, Biochemical Analyses, and FTCT Calculation

The FTCT was provided free of charge or with minimum costs to all participants by the Italian National Health Service as routine prenatal care. Ultrasound examination was performed by experienced obstetricians with the certificate of competence in the 11–13 weeks scan. The NT was measured in the mid-sagittal plane and neutral position of the fetus, with the crown rump length (CRL) ranging between 45 and 84 mm, according to the Fetal Medicine Foundation (FMF) criteria [25]. The maximum thickness of the subcutaneous translucency between the skin and the soft tissue overlying the cervical spine was measured. Both the nuchal translucency scan and biochemical analyses were carried out during the same day to be able to directly calculate the aneuploidy risk. Plasma glucose levels were measured by the hexokinase method, and free β-hCG and PAPP-A were assessed by the Immulite 2000 Analyzer (Siemens Healthcare Diagnostics GmbH, Eschborn, Germany), a random-access immunoassay analyzer with chemiluminescent detection, which provides reproducible results within 35 min (Intra-assay CV = ~3.5%, inter-assay CV = 6–8% for both analytes). The absolute levels of β-hCG and PAPP-A were converted to a multiple of the expected normal median (MoM), and adjusted for gestational age, maternal weight, ethnicity, previous malformations, and smoking status, before their inclusion in the risk algorithm. Free β-hCG values lower than 0.5 MoM, between 0.5 and 2.0 MoM, and higher than 2.0 MoM were categorized as low, normal, and high hCG levels, respectively [18]. The FTCT was calculated with PRISCA software version 4.0.15.9 (Siemens Healthcare Diagnostics GmbH, Eschborn, Germany). Screening test results were provided to the clinician as numerical information in relation to the patient’s age-related risk and a revised risk assessment based on age, serum analyte levels, and NT measurement.

### 2.3. Statistical Analysis

Initially, each quantitative trait was tested for normality using the Shapiro–Wilk normality test. The non-parametric Mann–Whitney test was used for comparisons of continuous variables, and the 2-tailed Fisher exact test for comparisons of proportions. Spearman’s rank correlation analysis was performed to explore the correlations between FTCT and its parameters with GDM. All significant variables were then forced into multivariable regression models. Logistic regression analysis was used to evaluate individual effects of each patient’s categorical FTCT parameter as a possible predictor of GDM, providing Odds Ratios (ORs) with 95% confidence bounds. Finally, receiver-operating-characteristic (ROC) analysis was performed to assess the discriminative capacity of any continuous trait in predicting GDM. In all analyses, statistical significance was fixed at an alpha level of 0.05. All calculations were performed with SPSS 20.0 software (SPSS Inc, Armonk, NY, USA).

## 3. Results

### 3.1. Characteristics of the Study Population

Figure 1 indicates how the study population was selected, whilst all clinical and biochemical features of women with and without GDM are shown in Table 1. Of the 2410 pregnant women that underwent screening for both FTCT and GDM, 596 (24.7%) were classified as GDM. As expected, in the GDM group, there was a higher percentage of women at high risk for GDM, as identified by the Italian guidelines (*p* < 0.001), even if only a small rate of them (36.1%) underwent the recommended early OGTT screening at 16–18 weeks of gestation (Table 1). In 42 cases, GDM was diagnosed by early screening, whereas in the remaining 554 patients, diagnosis was based on the conventional 24–28 weeks testing period (Table 1). Interestingly, diagnosis of GDM during OGTT was made by fasting glycemia in 64.4% of the patients, by 1 h post-load glycemia in 26.9% of them, and 2 h post-load in only 8.7% of the cases.

The Mann–Withney test revealed significant differences for age at diagnosis of GDM and pre-pregnancy body mass index (BMI) among the two groups, whereas Fisher’s exact test identified familial history of diabetes, previous diagnosis of GDM, previous diagnosis of PCOS, and number of pregnancies >1 as significant categorical traits (Table 1).

With respect to continuous parameters of the FTCT test, only the FTCT value was significantly different between women diagnosed with GDM and normal glucose tolerant women (*p* = 0.024) (Table 1). Among the categorical parameters of the FTCT test, we observed that a FTCT value <1:10000 was more frequent in women that developed GDM, in comparison to those unaffected (*p* = 0.01). Similarly, a PAPP-A value <1 MoM was more frequent among the GDM group in comparison to the unaffected group (*p* = 0.01) (Table 1)**.**

### 3.2. FTCT as Predictor of GDM

By using the Spearman univariate correlation analysis (Table 2), only the FTCT value was correlated with subsequent GDM diagnosis (*ρ* = 0.048, *p* = 0.018), whereas the other continuous FTCT parameters (β-hCG MoM, PAPP-A MoM, NT, and CRL) did not show significant correlations (Table 2). However, when we considered the glycemic values measured during OGTT, a correlation was observed for the FTCT value with fasting (*ρ* = 0.067, *p* = 0.001) and 1 h post-glucose load levels (*ρ* = 0.060, *p* = 0.004), and for PAPP-A MoM with fasting glucose levels (*ρ* = 0.082, *p* < 0.001) (Table 2), No other correlation was observed (Table 2).

Logistic regression analysis was then performed to identify the categorical conditions of the FTCT test that predict GDM. As shown in Table 3, a FTCT value <1:10000 appeared to be a good predictor of GDM (OR 1.26 (95% CI, 1.05–1.53), *p* = 0.016). Similarly, women with a PAPP-A value <1 MoM were at increased risk to develop GDM (OR 1.26 (95% CI, 1.05–1.53), *p* = 0.014) (Table 3), whereas a β-hCG value >2 MoM showed a protective effect against the risk of GDM (OR 0.66 (95% CI, 0.47–0.92), *p* = 0.014) (Table 3). Even an NT >1.5 mm was associated with GDM (OR 1.27 (95% CI, 1.01–1.60), *p* = 0.039), but this parameter failed to achieve statistical significance after adjustment for maternal age and pregravidic BMI (OR 1.20 (95% CI, 0.94–1.51), *p* = 0.140) (Table 3).

Finally, to verify whether any continuous FTCT parameter could be used as a specific predictor of GDM, we performed a ROC analysis. As indicated in Figure 2, among the continuous parameters tested for diagnosis of GDM, only the FTCT value was significant (*P* = 0.024), even if its accuracy was low (area under curve = 0.531 (0.504–0.557)) (Figure 2). All other traits tested were statistically not significant.

## 4. Discussion

FTCT is an effective tool to detect fetal chromosomal abnormalities. In recent years, its use has expanded worldwide, particularly in Western countries, not only for the rising rate of advanced maternal age pregnancies [26], but also for the increased awareness of the importance of FTCT in a larger number of pregnant women. In addition, several studies have been published, indicating that FTCT can provide useful information on the risk for the development of other pregnancy complications, such as pregnancy-induced hypertension, miscarriage, and impaired fetal growth [11,19,27].

Herein, we investigated the association of FTCT and its related serum biochemical parameters with GDM in Calabrian women, a Southern Italian population in which the high prevalence of obesity and T2DM is associated with a high rate of GDM (http://www.istat.it). Our recent studies, in this population, indicate that the proportion of women who would be diagnosed with GDM doubled (from 13 to 28%) after the introduction of IADPSG criteria [28], reaching a peak of about 50% among women at high risk for GDM, according to the current Italian guidelines [29].

After analysis of both continuous or dichotomous data (<1:10000 or not), we demonstrated that the FTCT value can be a predictor of GDM in early pregnancy, although with low accuracy. Even a PAPP-A value <1 MoM was predictive of GDM, whereas a β-hCG value >2 MoM was inversely associated with GDM. Our findings are consistent with a couple of previous studies [11,16], other than a recent systematic review and meta-analysis [30], indicating that both reduced first trimester levels of PAPP-A and free β-hCG are associated with the risk of GDM development. In addition, our data are consistent with those reported in a large Thailandian population, indicating a specific protective effect against GDM of a β-hCG value >2 MoM [18]. Besides, our results for PAPP-A are in agreement with a recent systematic review and meta-analysis indicating that PAPP-A has a low predictive accuracy overall [31]. Other studies, involving different populations, confirmed that first trimester PAPP-A levels are decreased in women who subsequently developed GDM, even though they did not report significant changes in β-hCG values [17,32,33]. Conversely, our results are inconsistent with three other reports in which first trimester serum-free β-hCG and PAPP-A were not significantly altered in pregnant women who subsequently developed GDM [19,20,34]. Although it is difficult to make direct comparisons, differences in methods used for detecting GDM and/or specific ethnic features of the study population may explain these discrepancies. In particular, two studies [17,33] employed a two-step approach with a 50 g nonfasting screen followed by 100 g OGTT with more conservative diagnostic cut-offs, when compared to the IADPSG criteria adopted in our study. In addition, the observed trend toward reductions of first trimester feto-placental biochemical markers, in pregnancies complicated by GDM, failed to reach statistical significance, presumably due to the low estimated prevalence of GDM within a Slovenian Caucasian population, using old diagnostic criteria (~2.3%) [19]. The impact of ethnicity in this context is also evident in a recent study [32] with a multi-ethnic cohort. Furthermore, while in some investigation, diagnosis of GDM was not defined according to IADPSG criteria [34], in other studies, the selection of GDM cases was based on a two-step approach, comprehensive of random glucose testing ≥6.7 mmol/L, followed by OGTT (WHO criteria) [20].

Consistent with previous reports, in our study, fetal NT was not associated with GDM [11,16,17,35].

A limitation of the present study, besides the retrospective design, is the lack of comparison with other pregnant populations. In addition, not all the women attending our Operative Unit for GDM screening had previously undergone FTCT. However, the prevalence of GDM and the class-risk rate were similar to those reported by us before [28,29,36,37], thus suggesting that a selection bias was very unlikely. In addition, the enrollment of pregnant women and the FTCT and OGTT tests were performed at a single center, and this, in our opinion, constitutes a strength of this work, which contributes to the minimization of the bias linked to inter-laboratory analytical variations [38]. Another strength of our study is the uniformity of GDM diagnosis among the study participants; in fact, we only enrolled women in whom screening for GDM was performed by adopting the new IADPSG cut-offs.

Overall, the interest in predicting the risk of GDM by simple and routinely applicable clinical models that can be used at first trimester is still ongoing [39]. FTCT is routinely implemented in the process of prenatal screening for chromosomal abnormalities, with the advantages of non-invasiveness and cost-effectiveness [6,40]. Although further prospective studies are needed to address this issue, stratification of GDM risk by FTCT tests may improve the detection rates for GDM of current selective screening approaches, exposing only high-risk women to the subjective and economic burden of 75 g OGTT. Lastly and most importantly, the assessment of GDM risk at first trimester would allow for early lifestyle changes and/or nutritional interventions [41] that may be effective in preventing the onset, or at least lessening the severity, of GDM, with significant health benefits for both mothers and babies.

## 5. Conclusions

Our findings indicate that FTCT is positively and significantly associated with GDM risk. Although its predictive significance is low, due to the clinical and public health relevance of GDM, our results confirm and extend previous observations, and suggest that FTCT and its related biochemical parameters have the potential to improve the efficacy of current selective screening strategies for GDM and target preventive care.

## Figures and Tables

**Figure 1 ijerph-16-03654-f001:**
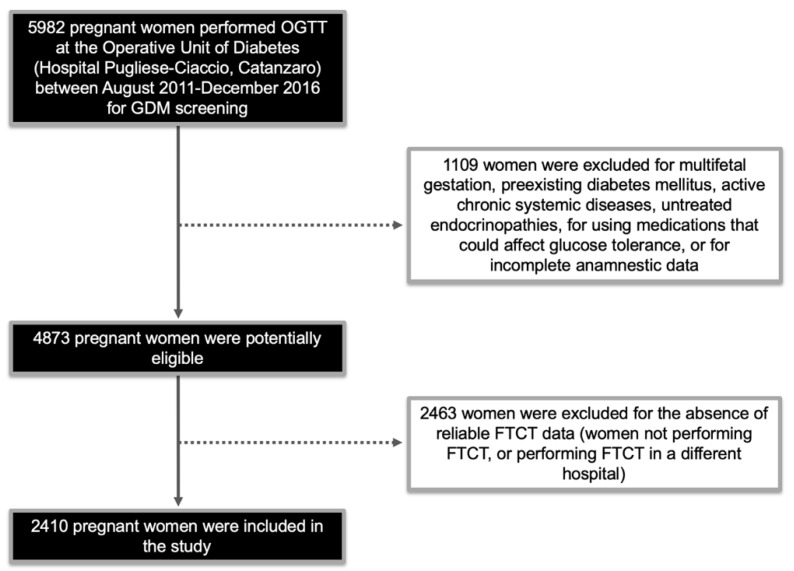
Flow chart indicating the selection of the study population. OGTT: oral glucose tolerance test; GDM: gestational diabetes mellitus; FTCT: first trimester combined test.

**Figure 2 ijerph-16-03654-f002:**
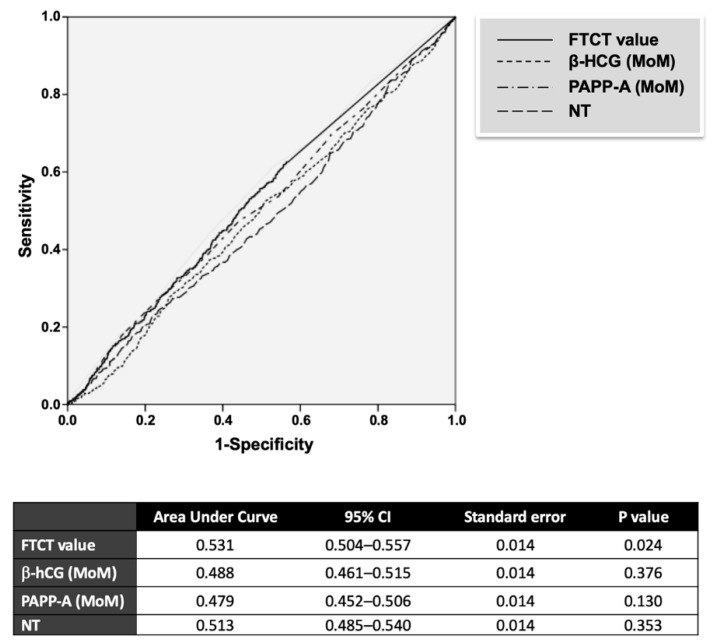
Receiver-operating-characteristic (ROC) curves of FTCT parameters for the prediction of GDM. β-hCG: β-human chorionic gonadotropin; PAPP-A: pregnancy-associated plasma protein A; NT: nuchal translucency.

**Table 1 ijerph-16-03654-t001:** Clinical and biochemical characteristics of the study population undergone first trimester combined test (FTCT) and gestational diabetes mellitus (GDM) screening tests.

Characteristics	No GDM (*N* = 1828)	GDM (*N* = 596)	*p* Value
Age, years	31 (28–34)	33 (29–34)	<0.001
Familiarity for T2DM, *N*	759 (41.5)	277 (46.5)	0.038
Pregravidic BMI, kg/m^2^	22.5 (21.7–25.5)	24.8 (21.9–28.7)	<0.001
Previous GDM, *N*	43 (2.4)	78 (13.1)	<0.001
Previous macrosomy, *N*	28 (1.5)	9 (1.5)	0.877
Smoking, *N*	75 (4.1)	24 (4.0)	0.970
PCOS, *N*	3 (0.2)	27 (4.5)	<0.001
Gravidity, *N*	1 (1–2)	2 (1–2)	0.282
No. of pregnancies >1	532 (29.1)	288 (48.3)	<0.001
High risk ^1^	150 (8.2)	169 (28.4)	<0.001
Intermediate risk ^1^	1126 (61.6)	270 (45.3)	<0.001
Low risk ^1^	552 (30.2)	157 (26.3)	<0.001
Early GDM screening, *N*	27 (18.0)	61 (36.1)	<0.001
GDM diagnosis at early screening	0	42 (68.9)	-
Later screening, *N*	1828	554	-
Time of FTCT, weeks	12.2 (11.5–12.5)	12.2 (11.5–12.5)	0.488
FTCT	0.00011 (0.00010–0.00023)	0.00013 (0.00010–0.00026)	0.024
β-hCG, MoM	0.91 (0.61–1.36)	1.02 (0.60–1.36)	0.376
PAPP-A, MoM	1.19 (0.82–1.67)	1.02 (0.77–1.68)	0.130
CRL, mm	58.7 (53.0–65.0)	59.0 (53.0–65.0)	0.363
NT, mm	1.10 (0.9–1.4)	1.10 (0.9–1.5)	0.352
FTCT <1:10000, *N*	1033 (56.5)	373 (62.6)	0.010
PAPP-A <1 MoM, *N*	680	257	0.011

Data are medians (interquartile range) or N (%). *p* Values refer to overall differences across groups as derived from the non-parametric Mann–Whitney test or Fisher’s exact test, respectively. T2DM: Type 2 diabetes mellitus; PCOS: polycystic ovary syndrome; FTCT: first trimester combined test; β-hCG: β-human chorionic gonadotropin; PAPP-A: pregnancy-associated plasma protein A; CRL: crown rump length; NT: nuchal translucency. ^1^ According to the Italian guidelines [23], high-risk women are those with at least one of the following parameters: Previous GDM, pre-pregnancy body mass index (BMI) ≥30 kg/m^2^, or fasting plasma glucose (FPG) at first trimester or before pregnancy between 100–125 mg/dL (5.6–6.9 mmol/L). For these women, early GDM screening at 16–18 weeks of gestation is recommended.

**Table 2 ijerph-16-03654-t002:** Univariate correlations between FTCT parameters and GDM and glycemic values during oral glucose tolerance test (OGTT).

Parameter	GDM (*N* = 596)	*p* Value	Glycemia Fasting	*p* Value	Glycemia 1 h-OGTT	*p* Value	Glycemia 2 h-OGTT	*p* Value
FTCT value	ρ = 0.048	0.018	ρ = 0.067	0.001	ρ = 0.060	0.004	ρ = −0.034	0.101
β-hCG MoM	ρ = −0.018	0.377	ρ = −0.035	0.091	ρ = 0.001	0.980	ρ = −0.004	0.844
PAPP-A MoM	ρ = −0.031	0.130	ρ = −0.082	<0.001	ρ = −0.018	0.401	ρ = −0.032	0.125
NT	ρ = 0.019	0.352	ρ = 0.015	0.466	ρ = 0.037	0.073	ρ = 0.029	0.161
CRL	ρ = 0.019	0.363	ρ = −0.010	0.626	ρ = 0.019	0.369	ρ = 0.035	0.096

Analysis was conducted with Spearman’s correlation. Correlation coefficient rho (*ρ*) is indicated.

**Table 3 ijerph-16-03654-t003:** Association of FTCT parameters with GDM.

Variable	No GDM	GDM	OR (95% CI)	*p* Value
FTCT <1:10000	1033/795	373/223	1.26 (1.05–1.53)	0.016
FTCT <1:8000	832/996	302/294	1.22 (1.02–1.47)	0.033
β-hCG <0.5 MoM	250/1578	99/497	1.25 (0.97–1.61)	0.089
β-hCG ≥2.0 MoM	208/1620	47/549	0.66 (0.47–0.92)	0.014
PAPP-A <0.4 MoM	44/1784	19/577	1.32 (0.77–2.29)	0.313
PAPP-A <1 MoM	680/1148	257/339	1.26 (1.05–1.53)	0.014
NT >1.5 mm	330/1498	130/466	1.27 (1.01–1.60)	0.039
NT >1.5 mm ^1^	330/1498	130/466	1.20 (0.94–1.51) ^1^	0.140 ^1^

Logistic regression analysis was performed to assess the independent role of each FTCT parameter on GDM. Odds ratio (OR) with confidence interval (CI) and P values are shown. ^1^ After adjustment for maternal age and pregravidic BMI.

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
