# Peer review of "First Trimester Combined Test (FTCT) as a Predictor of Gestational Diabetes Mellitus"

_ijerph, 2019, doi:10.3390/ijerph16193654_

Round 1

Reviewer 1 Report

This study is described in appropriate terms.  There is a significant unresolved issue: the apparent differences in earlier reports concerning the significance of the current paper.  Although the authors address this issue, the fact that there are these differences must be explained by analytic comparison between populations and conjointly conducted studies.

The presence of gestational diabetes in nearly 25% of the population appears high; should that category be further defined as to whether GDM was associated with pregnancy only; whether GDM was familial; etc.

In the Abstract, the word is "successful" not successive.

Author Response

Rev 1: This study is described in appropriate terms. There is a significant unresolved issue: the apparent differences in earlier reports concerning the significance of the current paper.  Although the authors address this issue, the fact that there are these differences must be explained by analytic comparison between populations and conjointly conducted studies.

AUTHORS: As requested, in the revised version of the manuscript, we now provide an analytic description of the reasons that can explain discrepancies with other studies (lines 218-229).

Rev 1: The presence of gestational diabetes in nearly 25% of the population appears high; should that category be further defined as to whether GDM was associated with pregnancy only; whether GDM was familial; etc.

AUTHORS: We now refer to the epidemiological features of GDM in Calabrian population (lines 198-203).

Rev 1: In the Abstract, the word is "successful" not successive.

AUTHORS: The word "successive" has been replaced by the more appropriate term "subsequent".

Reviewer 2 Report

Overall, I think this retrospective analysis is a valuable addition to the literature, as it investigates a topic that holds important implications for gestational diabetes mellitus diagnosis.

However, I have several concerns about how the discussion is presented.

The presented results are partially conflicting with other studies (lines 209-216). In the Authors opinion it is possible that the discrepancies may be due, at least in part, to differences in study design (e.g. GDM screening methods, markers detection method used, etc.), ethnic and population-specific features, and diversity in the prevalence of GDM. In my opinion, this sentence only suggests a direction of a possible discussion, without the detailed analysis of the problem.

Please, consider re-working your discussion to contain essential details of how you interpret the novel findings from your study and how they might be of clinical relevance.

Author Response

Rev 2: The presented results are partially conflicting with other studies (lines 209-216). In the Authors opinion it is possible that the discrepancies may be due, at least in part, to differences in study design (e.g. GDM screening methods, markers detection method used, etc.), ethnic and population-specific features, and diversity in the prevalence of GDM. In my opinion, this sentence only suggests a direction of a possible discussion, without the detailed analysis of the problem.

AUTHORS: As requested also by reviewer 1, this concern has been addressed in the revised version of the manuscript (lines 218-229).

Rev 2: Please, consider re-working your discussion to contain essential details of how you interpret the novel findings from your study and how they might be of clinical relevance.

AUTHORS: To accomplish the reviewer’s concern, we now refer to this point (see lines 242-250 in the Discussion section).

Reviewer 3 Report

Abstract

Is clear concise and very effective

Line 47- Surely the IADSPG have not actually caused the increase in diabetes but rather altering their cut off point has increased the number of women diagnosed with GDM- needs a slight re-wording.

The paper is very well written throughout and no doubt highly valuable data for the early detection of GDM- my recommendation is to accept- with only one slight amendment which I make above.

Author Response

Rev 3: Line 47- Surely the IADSPG have not actually caused the increase in diabetes but rather altering their cut off point has increased the number of women diagnosed with GDM- needs a slight re-wording.

AUTHORS: Sentence at line 47 has been slightly modified